# Design of a Fuzzy Adaptive Voltage Controller for a Nonlinear Polymer Electrolyte Membrane Fuel Cell with an Unknown Dynamical System

Reza Ghasemi [1], Mehdi Sedighi [2], Mostafa Ghasemi [3],[*] and Bita Sadat Ghazanfarpoor [4]

1 Department of Electrical Engineering, University of Qom, Qom 3716146611, Iran; r.ghasemi@qom.ac.ir
2 Department of Chemical Engineering, University of Qom, Qom 3716146611, Iran; sedighi@qom.ac.ir
3 Chemical Engineering Section, Faculty of Engineering, Sohar University, Sohar 311, Oman
4 Engineering Department, Islamic Azad University (IAU), Damavand Branch, Tehran 1477893855, Iran; b.ghazanfarpour@gmail.com
* Correspondence: mbaboli@su.edu.om or mostafghasemi@gmail.com

**Abstract:** This paper presents a fuzzy adaptive controller (FAC) for improving the efficiency and stability of fuel cells, assuming that the nonlinear dynamic model of the system is unknown. In polymer electrolyte membrane fuel cells, the output voltage should be controlled within a given interval. In contrast to prior studies that focused on designing controllers for known dynamical models of PEM fuel cells, the suggested approach addresses the real-world case of a PEM fuel cell with unknown dynamics. An intelligent technique is identified in the suggested strategy to approximate the state-space model of fuel cells to manage unknown functions. On an unknown model of fuel cells, traditional adaptive and fuzzy adaptive controllers are both implemented and compared. The main advantages of the proposed methodology are (1) stability of the closed-loop system using Lyapunov, (2) robustness against external disturbances, (3) application of the FAC to a PEM fuel cell, (4) convergence of the tracking error to 0, and (5) overcoming both unknown dynamics and uncertainty in the system. The most important and valuable advantages of the proposed system are its robustness, tracking error convergence, and Lyapunov stability. This manuscript aims to illustrate the responsiveness and fluency of the proposed procedure using a mathematical formulation of a multi-quadrotor system. As a result, the FAC is more efficient than the traditional one. To validate the controller performance, both the adaptive and fuzzy adaptive controllers are applied to a numerical model of a fuel cell and then compared.

**Keywords:** adaptive control; fuzzy adaptive control; Lyapunov stability; polymer electrolyte membrane fuel cells (PEMFCs); simulation

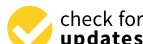



## 1. Introduction

Energy is one of the most important subjects globally and is essential to all the processes that happen in the world. Conventional energy generation methods, such as burning fossil fuels, cause environmental difficulties, one of which is an increase in greenhouse gas emissions, which contributes to climate warming [1]. Natural gas, petroleum, and coal are the principal (and nonrenewable) sources of electricity and contributors to environmental pollution of the air and environment, threatening health [2]. Renewable Energy Sources (RESs) have attracted more attention, and extensive studies have been conducted to develop alternative electricity generation methods to address the global warming crisis [3]. The Paris Agreement's sustainable development goal concept of zero carbon emissions is the foundation of decarbonization implemented in the majority of developed countries worldwide. One of the efforts being made to decarbonize the environment is the use of hydrogen fuel cell technology. Fuel cells are one of the most enticing energy sources considered as an alternative type of energy [4,5]. Compared to combustion engines, fuel cells emit little or

no pollution. Because there are no carbon dioxide emissions from hydrogen fuel cells, they address critical climate challenges. There are no air contaminants that produce pollution or health problems at the point of operation. Fuel cells run silently because they have few moving parts. The increasing demand for energy resources has recently widened the field of research into clean energy devices, particularly fuel cells and sustainable energy systems. The development of fuel cells and sustainable energy systems is an important method for improving clean energy efficiency in the future, particularly in the fields of electric vehicles and machines, clean power generation for industries and citizens, cogeneration, and so on. The most recent scientific research findings and R&D trends in fuel cells and sustainable energy systems require the continuing collaboration of researchers from various backgrounds to encourage the rapid growth of science and technology in fuel cells and sustainable energy systems. Fuel cells function similarly to batteries, but they do not need to be recharged. They generate electricity and heat as long as fuel is available. A fuel cell is made up of two electrodes, one negative (or anode) and one positive (or cathode), sandwiched around an electrolyte. The anode receives fuel, such as hydrogen, while the cathode receives air. A catalyst at the anode of a hydrogen fuel cell separates hydrogen molecules into protons and electrons, which travel in opposite directions to the cathode. Electrons flow through an external circuit, causing electricity to flow. Protons pass through the electrolyte and arrive at the cathode, where they mix with oxygen and electrons to produce water and heat. Polymer electrolyte membrane fuel cells (PEMFCs) offer many advantages compared with other fuel cells, including high power density, low temperature, and fast response. However, PEMFCs have several shortcomings, such as low reliability, short duration, and high cost, which limit their applications [6,7]. Fuel cells are used in a wide range of industrial and nonindustrial applications, including in airports, schools, workplaces, hospitals, and hotels. The PEMFC is a good stationary-state energy source, but it cannot respond quickly to load variations. Power problems result in extreme load fluctuations and load current variations. Moreover, fuel cells are challenged by voltage fluctuations, which can be reduced by applying suitable control strategies and power converters in the system [8].

Although the proton exchange membrane fuel cell (PEMFC) continues to pique R&D interest due to its high energy density, its commercialization is hampered by several challenges, including cost reduction, improved performance, and increased durability with time. While they can be mitigated by some choices such as material selection, voltage reversals and fuel starvation also have an impact on PEMFC durability, as well as performance. In this paper, the reaction, thermal, water management, and power electronic subsystems of PEMFCs are critically studied and reviewed, with a focus on control strategies to avoid fuel starvation. Proportional–Integral–Derivative (PID) controllers, which manipulate hydrogen and air flow rates, are often employed in feedback voltage control and feedforward current control. Self-tuning PID controllers or sliding mode controllers respond quickly to changing dynamics. Adaptive controllers (ACs), such as load governors and extremum-seeking controllers, continuously update control action. Model predictive control (MPC) predicts system behavior and updates the controller action using a PEMFC model. Recently, artificial intelligence (AI) techniques such as neural network control (NNC), fuzzy logic control (FLC), and FLC-PID control have been used in PEMFC system control because they are simpler and less expensive to implement than the AC and MPC and also produce better results. Fuel cells are used in many applications, such as grid-connected and stand-alone settings. A power conditioning unit is necessary to match the load and fuel cell.

The fuel cell is viewed and controlled as an exclusive application across all given methodologies that have been presented. PEMFCs are expected to have a lifetime of 40,000 h and an electrical efficiency of 40–50%. PEMFCs provide greater power density compared to other types of fuel cells. They can produce more power than other fuel cells with the same size, volume, and weight. In recent years, a large amount of research has been conducted on FACs. In Ref. [9], the affine of nonlinear systems was modeled alongside a stable controller design based on TS. In multi-variable Takagi–Sugeno fuzzy

systems, a sliding mode FAC has been devised. Ref. [10] provides a fuzzy model reference state tracking controller based on canonical large-scale nonlinear systems. However, the interaction is viewed as a bounded disturbance with inaccessible states.

Finally, an adaptive controller is designed with high efficiency, better performance, rapid convergence, and robustness. As a result, the output voltage can be adjusted by adjusting the hydrogen flow of the proposed controller. The nonlinear model of PEM fuel cells is controlled using conventional sliding mode controllers, which ensures stability and reduces chattering [11]. Model-based optimal control and the efficient online stochastic estimation of fuel cell parameters were previously developed for PEM fuel cells [12]. In [13], an optimal linear parameter varying (LPV) controller was developed for nonlinear PEM fuel cells using a systematic design method based on linear matrix inequality. The authors of [14] developed a robust controller for the linear model of PEMFC in the presence of linear disturbances [15], concentrating on designing a decentralized PID controller based on PSO to apply to PEMFCs. The authors of [16] discussed L1 norm fast model predictive control for linearized PEM fuel cell system models. The authors of [17] focused on developing both fuzzy and adaptive PID controllers in order to regulate the Tramway system's PEM fuel cell methods. In [18], a functional order fuzzy PID controller was developed for a fuel cell, utilizing a neural network optimization approach to update the PID parameters [18]. Authors in [19] concentrate on super-twisting sliding mode and adaptive ANFIS procedure to increase stack life of the PEMFC. Ref. [20] deals with robust fault tolerant controller based on perturbation theory for a class nonlinear model of PEMFC. The wide researches on fault diagnosis on PEM Fuel cell are investigated in [21]. Classic sliding mode controller is designated for a linear class of PEM fuel cell [22].

Previous research has focused on either affine nonlinear systems or fuzzy systems to approximate the unknown nonlinear functions of the PEM fuel cell, which has resulted in both system structural limitations and increased computational volume. This paper presents a method for approximating controllers instead of nonlinear functions to reduce the computational volume. The nonaffine, nonlinear PEM fuel cell is designated as a universal form of the nonlinear system in this approach.

This manuscript designs an adaptive controller for the LTI model of a PEM fuel cell in the presence of both unknown dynamical models and uncertainties. In addition, using experts' knowledge, challenging both the uncertainties and unknown models of fuel cells, a fuzzy adaptive controller is developed. Section 2 discusses fuel cell analyses, such as basic fuel cell operation, open circuit fuel cell voltages, and small-signal fuel cell models. Section 3 introduces the novel adaptive and fuzzy adaptive controllers, as well as the required proofs. Section 4 uses simulations to demonstrate the theoretical results. Finally, the results are summarized in Section 5.

## 2. Fuel Cell Concepts and Model

The following sections are categorized into three subheadings. The equations for a single PEMFC are presented in Section 2.1. The equations for the open-circuit voltage of the fuel cell are given in Section 2.2. Finally, in Section 2.3, fuel cell small-signal models are examined in more detail ([4,5] and, furthermore, in [6]).

### 2.1. Operating Principle of the Fuel Cell

The fuel cell is made up of porous catalyzed electrodes divided by a solid electrolyte that contains a single polymer electrolyte membrane made of an ion-conduction polymer. A predetermined pressure level feeds the hydrogen-rich gas from the cell's anode side, which subsequently diffuses over porous electrodes until it enters the anode. In the catalytic layer, where it reacts, protons are produced and electrons are released. An external circuit is used to transfer electrons released at the anode to the cathode, and two $H^+$ ions flow to the cathode simultaneously through the membrane. At the cathode, the air is stored. In the

catalytic layer, oxygen reacts with the electrons from the $H^+$ ions. The PEMFC's anode and cathode chemical reactions are described in (1).

$$\begin{cases} H_2 \rightarrow 2H^+ + 2e & (Anode) \\ 2H^+ + \frac{1}{2}O_2 + 2e \rightarrow H_2O & (Cathode) \end{cases} \tag{1}$$

As a result, the total cell reaction is determined by

$$H^+ + \frac{1}{2}O_2 \rightarrow H_2O \quad (Overall\ Reaction) \tag{2}$$

The PEMFC's overall scheme is shown in Figure 1.

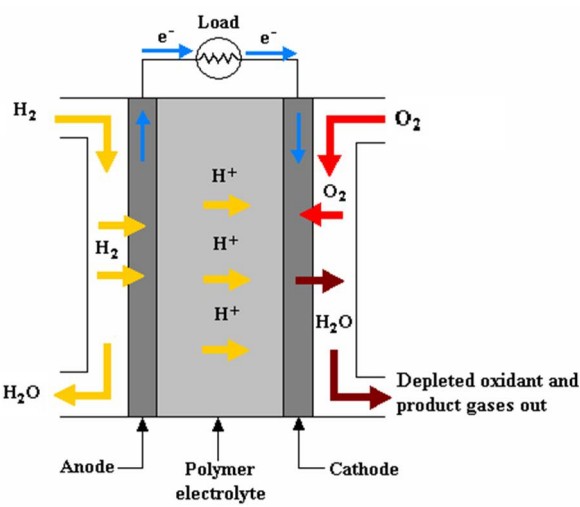

**Figure 1.** The electrochemical reaction taking place in the PEMFC [23].

### 2.2. Open Circuit Voltage of PFMFCs

Fuel cells convert chemical energy directly into electricity. They generate chemical energy according to the change in the Gibbs Free Energy (GFE) ($\Delta G_f$), which equals the difference between the GFE of the reactants and the GFE of the product. As shown below, the GFE represents the overall chemical reaction in the fuel cell.

$$\Delta G_f = G_f(products) - G_f(reactants) = -\left(G_f\right)_{H_2O}\left[+\left(G_f\right)_{H_2} + \left(G_f\right)_{O_2}\right] \tag{3}$$

The GFE changes with temperature and pressure, which can be expressed as

$$\Delta G_f = \Delta G_f^0 - \overline{R}T_{fc}ln\left(P_{H_2}P_{O_2}^{\frac{1}{2}} \middle/ P_{H_2O}\right) \tag{4}$$

The electrical objective in a reversible system is to pass two moles of electrons around the external circuit equivalent to the GFE change. As a result, the reversible voltage of a fuel cell can be calculated using Equation (3) as follows:

$$\Delta G_f = -2FE \tag{5}$$

where

$$E = \frac{-\Delta G_f}{2F} = \frac{-\Delta G_f^0}{2F} + \frac{\overline{R}T_{fc}}{2F}ln\left(P_{H_2}P_{O_2}^{\frac{1}{2}} \middle/ P_{H_2O}\right) \tag{6}$$

In Equation (6), the voltage, $E$, is referred to as the reversible open-circuit voltage or "Nernst" voltage of a hydrogen fuel cell. In most cases, fuel cell reactions are irreversible, and some chemical energy is transferred to heat and pulses until the fuel cell voltage, $V_{fc}$,

falls below the value specified in Equation (6). According to [24], the following Nernst equation describes the reversible voltage of the cell:

$$E = 1.229 - 0.85 \times 10^{-3}\left(T_{fc} - 298.15\right) + 4.3085 \times 10^{-5} T_{fc}[ln(P_{H_2}) + \frac{1}{2}ln(P_{O_2})] \quad (7)$$

$V_{Loss}$, or voltage loss, is made up of three voltages:

$$V_{Loss} = V_{activ.} + V_\Omega + V_{conc.} \quad (8)$$

The voltage drop $(V_{activ.})$ that impacts the cathode and anode is referred to as the activation overpotential. $V_\Omega$ denotes the ohmic voltage drop associated with the electrodes and resistances of the conducting protons, which is used to calculate the ohmic voltage. Reducing the concentration of reactant gases results in a voltage drop that is represented by $V_{conc.}$. Alternatively, there is a concentration overpotential due to the mass transport of hydrogen and oxygen. Equation (8) can be rewritten as follows according to [25]:

$$V_{Loss} = r(i + i_n) + aln\left(\frac{i + i_n}{i_0}\right) - bln\left(1 - \frac{i + i_n}{i_l}\right) \quad (9)$$

The following equation can represent a single cell's output voltage:

$$V_{fc} = E_{Nernst} - (V_{activ.} + V_\Omega + V_{conc.}) \quad (10)$$

which is described in [8,26], as below:

$$V = N(E^0 + \frac{RT}{2F}ln\left(P_{H_2}P_{O_2}^{\frac{1}{2}} \Big/ P_{H_2O}\right) - L) \quad (11)$$

*2.3. Fuel Cell Small-Signal Models*

In this paper, we develop dynamic and nonlinear models based on the fuel cell models presented in [27]. Nonlinear terms are included in Equations (9) and (11). To linearize the cell voltage, a small-perturbation method must be used for the fuel cell dynamics model as an almost linear system. As a result, at these operating points, the changing behavior of the cell's output voltage can be estimated extremely quickly for modest changes in the input variables. In [27], the authors proposed the following linear state-space model. Using $\Delta x = [\Delta P_{H_2} \Delta P_{O_2} \Delta P_{H_2O_C}]^T, \Delta y = \Delta E, \text{and} \Delta u = [\Delta H_{2_{in}} \Delta O_{2_{in}} \Delta H_2O_{C_{in}} \Delta i]^T$, the model's parameters are defined as below:

$$\Delta \dot{x} = A\Delta x + B\Delta u$$

$$\Delta y = C\Delta x + D\Delta u, \quad (12)$$

The three system states for $\Delta x$ in Equation (12) represent perturbations of the partial pressures of $H_2$, water, and oxygen vapor inside the cells. In addition, the four inputs for $\Delta u$ indicate changes in the inlet flow rates of water, oxygen, and hydrogen vapor and the output current density. Furthermore, a change in the fuel cell stack voltage is indicated by the system output, y. The proposed controller is designed using matrices A, B, C, and D, which are based on the linearization of the nonlinear system in Ref. [27].

## 3. Adaptive Controller Design for PEMFCs

This section describes the proposed adaptive controller (AC) and fuzzy adaptive controller (FAC). This strategy is intended to find a process for improving the effectiveness of the FAC from the AC. The necessary proofs for the adaptive and fuzzy adaptive controllers are presented in Sections 3.1 and 3.2, while the necessary extended proofs for the fuzzy adaptive controller are presented in Section 3.3.

### 3.1. Design of the Proposed Adaptive Controller

To compensate for characteristic changes in the controlled process, adaptive control systems synthesize adaptive gains. Several types of adaptive control systems have been developed, each of which adjusts the controller's parameters in a different way. Model reference adaptive control (MRAC) seeks to resolve performance characteristics. The exact model is as follows:

$$\Delta \dot{x} = A\Delta x + B\Delta u \tag{13}$$

In addition, the reference model is

$$\dot{x}_m = A_m X_m + B_m r \tag{14}$$

The general linear control law is as follows:

$$u = Mr - Lx \tag{15}$$

Based on the two matrices $M$ and $L$, Equation (34) can be substituted into Equation (32) to derive the following equation:

$$\dot{x} = Ax + B(Mr - Lx) = (A - BL)x + BMr \tag{16}$$

The tracking error vector can be defined as follows:

$$e = x - x_m \tag{17}$$

Equations (14) and (16) can be replaced by (17) to give $\dot{e} = \dot{x} - \dot{x}_m$. After some mathematical manipulation, we obtain

$$\dot{e} = A_m e + \left( \overbrace{\underbrace{A - BL}_{\widetilde{A}} - A_m}^{A_C} \right) x + \left( \underbrace{BM - B_m}_{\widetilde{B}} \right) \tag{18}$$

The following candidate Lyapunov function can be used to determine the system's stability:

$$V = \frac{1}{2} e^T P_1 e + \frac{1}{2} tr \left( \widetilde{A}^T P_2 \widetilde{A} + \widetilde{B}^T P_3 \widetilde{B} \right) \tag{19}$$

$P_1$, $P_2$, and $P_3$ are positive definite matrices, and tr denotes the trace of a matrix. The following is the Lyapunov function's time derivative:

$$\dot{V} = \frac{1}{2} \dot{e}^T P_1 e + \frac{1}{2} e^T P_1 \dot{e} + \frac{1}{2} tr \left( \dot{A}_C^T P_2 \widetilde{A} + \widetilde{A}^T P_2 \dot{A}_C + \dot{B}_C^T P_3 \widetilde{B} + \widetilde{B}^T P_3 \dot{B}_C \right) \tag{20}$$

This can be rewritten as follows:

$$\dot{V} = \frac{1}{2} \left( e^T A_m^T + x^T \widetilde{A}^T + r^T \widetilde{B} \right) P_1 e + \frac{1}{2} e^T P_1 \left( A_m e + x + \widetilde{B}.r \right)$$
$$+ tr \left( \widetilde{A}^T P_2 \dot{A}_{C+} \widetilde{B}^T P_3 \dot{B}_C \right) \tag{21}$$

Matrix $Q_1$ can be considered as follows:

$$\dot{V} = \frac{1}{2} e^T \left( \underbrace{A_m^T P_1 + P_1 A_m}_{-Q} \right) e + \frac{1}{2} x^T \widetilde{A} P_1 e + \frac{1}{2} r^T \widetilde{B} P_1 e + \frac{1}{2} e^T P_1 \widetilde{A} x + \frac{1}{2} e^T P_1 \widetilde{B} r$$
$$+ tr (\widetilde{A}^T P_2 \dot{A}_C + B^T P_3 \dot{B}_C) \tag{22}$$

This can be rewritten as

$$\dot{V} = -\frac{1}{2}e^T Q_1 e + x^T \widetilde{A} P_1 e + r^T \widetilde{B} P_1 e + tr\left(\widetilde{A}^T P_2 \dot{A}_C + \widetilde{B}^T P_3 \dot{B}_C\right) \tag{23}$$

where $x^T \widetilde{A} P_1 e \equiv tr\left(\widetilde{A} P_1 e x^T\right)$ and $r^T \widetilde{B} P_1 e \equiv tr(\widetilde{B} P_1 e r^T)$. Equation (23) can then be written as follows:

$$\dot{V} = -\frac{1}{2}e^T Q_1 e + tr(\widetilde{A}\left(P_1 e x^T + P_2 \dot{A}_C\right) + \widetilde{B}\left(P_3 \dot{B}_C + P_1 e r^T\right)) \tag{24}$$

For the system to be stable, the following conditions must be met:

$$\dot{A}_c = -P_2^{-1} P_1 e x^T, \quad \dot{B}_c = -P_3^{-1} P_1 e r^T \tag{25}$$

Equation (16) leads to the following equation:

$$\dot{A}_C = -BL, \quad \dot{B}_C = B\dot{M} \tag{26}$$

Equation (25) is replaced by Equation (26) as follows:

$$B\dot{L} = P_2^{-1} P_1 e x^T, \quad B\dot{M} = -P_3^{-1} P_1 e r^T \tag{27}$$

The following updated laws can then be derived:

$$\dot{L} = B^+ P_2^{-1} P_1 e x^T, \quad \dot{M} = -B^+ P_3^{-1} P_1 e r^T \tag{28}$$

Equation (15) can be used as a reliable input for the system by substituting Equation (28) with the control law.

### 3.2. Fuzzy Adaptive Controller Design for PEMFCs

A fuzzy system is developed using an adaptive approximation of the unknown ideal controller. The following equation is to be considered:

$$\begin{cases} \dot{x} = Ax + Bu \\ y = C^T x + Du \end{cases} \tag{29}$$

As a result, the proposed model's control objective is to create an FAC for the system given by Equation (29) so that the system's output follows the appropriate trajectory for all closed-loop signals to remain limited. The following is the tracking error vector:

$$e = y - y_m \tag{30}$$

Taking the derivative of both sides of Equation (30) and applying Equation (29) results in the following equation:

$$\dot{e} = \dot{y} - \dot{y}_m = C^T \dot{x} + D^T \dot{u} - \dot{y}_m = C^T Ax + C^T Bu + D^T \dot{u} \tag{31}$$

Considering $u^*$ as an ideal input of this system, and then adding and subtracting the term $C^T Bu^*$ from $C^T Ax + C^T Bu$, leads to

$$C^T Ax + C^T Bu = C^T Ax + C^T Bu - C^T Bu^* + C^T Bu^* \tag{32}$$

Using $e_u = u - u^*$ and rewriting Equation (32), the following equation is determined:

$$C^T Ax + C^T Bu^* + C^T B\left(u - u^*\right) = C^T Ax + C^T Bu^* + C^T Be_u \tag{33}$$

We assume that $\underline{v} = [V_1 \ldots V_n]^T$, which is defined as follows:

$$v = Ke + \dot{y}_m + v' \tag{34}$$

$K$ is set in such a way that $\dot{e} = Ke$ is asymptotically stable, and $v'$ is defined later in the update rules in Equation (49) as pseudo-control input that meets the following condition:

$$C^T A x + C^T B u - v = 0 \tag{35}$$

The following result is obtained by substituting Equation (34) with Equation (35), and subsequently with Equation (31).

$$\dot{e} = \underbrace{C^T A x + C^T B u^* - v} + C^T B e_u - \dot{y}_u + v \tag{36}$$

The replacement of Equations (34) and (35) with Equation (36) yields

$$\dot{e} = ke + \dot{y}_m + v' + \underbrace{C^T B}_{M} e_u = ke + v' + M e_u + \dot{y}_m \tag{37}$$

*3.3. Fuzzy Adaptive Controller Design*

There is an ideal controller that meets the control objectives described in the previous subsection. A set of if–then rules comprises the fuzzy rule base. Assume that there are M-many rules, with the *l*th rule being

$$Rule^l : \ if(x_1 \ is \ A_1^l \ \ldots \ x_n \ is \ A_1^l) \ then \ (y \ is \ B^l) \ \ l = 1, 2, \ldots, M$$

The fuzzy inference transfers fuzzy sets in U to fuzzy sets in V using if–then rules from the fuzzy rule base. This section describes how to develop a fuzzy system using adaptive approximation. The optimal controller is denoted by

$$u^* = f(z) = w^T \theta^* \tag{38}$$

where $z = [x, v]$ and $\theta = [y^1 y^2 \ldots y^M]$ are the vectors including all consequent parameters, and $w(x) = [w_1(x) w_2(x) \ldots w_M(x)]^T$ is a set of fuzzy basis functions:

$$w_i(x) = \frac{\prod_{i=1}^n \mu_{A_i^l}(x_i)}{\sum_{l=1}^M \prod_{i=1}^n \mu_{A_i^l}(x_i)} \tag{39}$$

Assume that $u^*$ and $\theta^*$ are estimated as $u_T$ and $\theta$, respectively, and that $u_r$ is a robust controller to compensate for uncertainties, disturbances, and approximation errors. Rewriting Equation (38) can lead to the following equation:

$$u_T = w^T \theta + u_r \tag{40}$$

As $e_u = u - u^*$, using Equation (40) results in

$$e_u = u - u^* = w^T \underbrace{\left(\theta - \theta^*\right)}_{\tilde{\theta}} + u_r = w^T \tilde{\theta} + u_r \tag{41}$$

Equation (41) leads to the following equation for the error derivation:

$$\dot{e} = Ke + \dot{v} + M w^T \tilde{\theta} + M u_r + \dot{y}_m \tag{42}$$

Observe the following candidate for the Lyapunov function:

$$V = \frac{1}{2}e^T P e + tr\left(\frac{\tilde{\theta}^T \tilde{\theta}}{2\gamma_1} + \frac{v'^T v'}{2\gamma_2} + \frac{u_r^T u_r}{2\gamma_3}\right) \tag{43}$$

where $\tilde{\theta} = \theta - \theta^*$, $\gamma_1, \gamma_2, \gamma_3 > 0$, and $P$ is a positive definite matrix. The time derivative of this Lyapunov function is

$$\dot{V} = \frac{1}{2}\dot{e}^T P e + \frac{1}{2}e^T P \dot{e} + tr\left(\frac{\dot{\tilde{\theta}}^T \tilde{\theta}}{2\gamma_1} + \frac{\tilde{\theta}^T \dot{\theta}}{2\gamma_1} + \frac{\dot{v}'^T v'}{2\gamma_2} + \frac{v'^T \dot{v}'}{2\gamma_2} + \frac{\dot{u}_r^T u_r}{2\gamma_3}\right) \tag{44}$$

where $\tilde{\theta} = \theta - \theta^*$, $\dot{\tilde{\theta}} = \dot{\theta}$, and furthermore, $\theta$ indicates unknown parameters that will be defined later. Using Equation (42), Equation (44) leads to

$$\begin{aligned}\dot{V} = &\tfrac{1}{2}e^T k^T P e + \tfrac{1}{2}v'^T P e + \tfrac{1}{2}\tilde{\theta}^T w M^T P e + \tfrac{1}{2}u_r^T P e + \tfrac{1}{2}e^T P k e + \tfrac{1}{2}e^T P v' \\ &+ \tfrac{1}{2}e^T P M w^T \tilde{\theta} + \tfrac{1}{2}e^T P u_r + e^T P \dot{y}_m + tr\left(\frac{\dot{\tilde{\theta}}^T \tilde{\theta}}{\gamma_1} + \frac{\dot{v}'^T v'}{\gamma_2} + \frac{\dot{u}_r^T u_r}{\gamma_3}\right)\end{aligned} \tag{45}$$

Equation (45) can be modified as follows:

$$\begin{aligned}\dot{V} = &\tfrac{1}{2}e^T \underbrace{\left(K^T P + P K\right)}_{-Q} + v'^T P e + \tilde{\theta}^T w M^T P e + u_r^T P e + e^T P \dot{y}_m + tr\left(\frac{\dot{\tilde{\theta}}^T \tilde{\theta}}{\gamma_1}\right. \\ &\left. + \frac{\dot{v}'^T v'}{\gamma_2} + \frac{\dot{u}_r^T u_r}{\gamma_3}\right)\end{aligned} \tag{46}$$

For any positive definite symmetric matrix $Q$, there exists a unique positive definite symmetric solution $P$ in the following Lyapunov equation:

$$K^T P + P K = -Q \tag{47}$$

Equation (47) is substituted into Equation (46) to give

$$\dot{V} = -\frac{1}{2}e^T Q e + e^T P \dot{y}_m + tr(P e v' + w M^T P e \tilde{\theta} + P e u_r + \frac{\dot{\tilde{\theta}}^T \tilde{\theta}}{\gamma_1} + \frac{\dot{v}'^T v'}{\gamma_2} + \frac{\dot{u}_r^T u_r}{\gamma_3}) \tag{48}$$

The values tr(.) must be zero in order to achieve zero tracking error in the closed-loop system. The following equations must also hold:

$$\dot{v}' = -\gamma_2 e^T P^T$$
$$\dot{u}_r = -\gamma_3 e^T P^T$$

$$\dot{\theta} = -\gamma_1 e^T P^T M w^T \tag{49}$$

The compact $\Omega = \left\{e \big| \|e\| < \|Q\| \big/ \|P \dot{y}_m\| \right\}$ ensures the stability of the closed-loop system. The proof is thus complete.

## 4. Simulation Confirmation

This section illustrates the results of applying the proposed adaptive controller and fuzzy adaptive controller to a PEMFC system.

### 4.1. Adaptive Controller Simulations

Table 1 presents the parameters for the PEMFC system [2]. When these criteria are followed, the system is controllable and observable, and the control goal is to make the system's state follow a desired trajectory. The simulation and modeling of the PEMFC system were validated using MATLAB. The control inputs for the adaptive control system under study are depicted in Figures 2–5.

**Table 1.** PEMFC system parameters.

| Parameter | Value | Parameter | Value |
|---|---|---|---|
| $N$ | 32 | $V_A$ | 6.495 [cm$^3$] |
| $T$ | 338.5 [K] | $V_C$ | 12.96 [cm$^3$] |
| $pH_2$ | 40.28 [kPa] | $P_{OP}$ | 101 × 103 [Pa] |
| $pO_2$ | 19.28 [kPa] | $I$ | 2 [A] |
| $pH_2O_a$ | 10.28 [kPa] | $i$ | 0.4 [A/m$^2$] |
| $R$ | 8.31 [J/K·mol] | $i_n$ | 0.2 [A] |
| $F$ | 96,439 [C/mol] | $i_0$ | 0.2 [A] |
| $i_l$ | 1 [A] | $r$ | 0.0635 [Ω] |
| $V$ | 1.6 [V] | | |

The following are the reference model parameters for the first scenario:

$$A = \begin{bmatrix} -1 & 0 & 0 & 0 \\ 0 & -2 & 0 & 0 \\ 0 & 0 & -3 & 0 \\ 0 & -1 & 0 & 0 \end{bmatrix}, B = \begin{bmatrix} 1 & 0 & -1 & 0 \\ 0 & 1 & 0 & 0 \\ 0 & 0 & 1 & 1 \\ 0 & 1 & 0 & 0 \end{bmatrix}$$

$$C = \begin{bmatrix} 1 & 0.5 & -2 & 0 \end{bmatrix}, D = \begin{bmatrix} 0 & 0 & 0 & 0 \end{bmatrix} \tag{50}$$

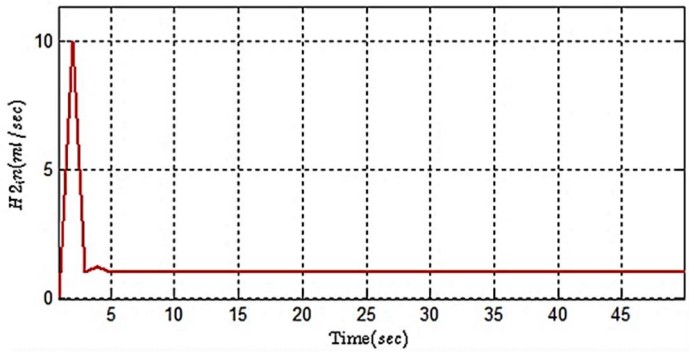

**Figure 2.** The first control input for the adaptive controller.

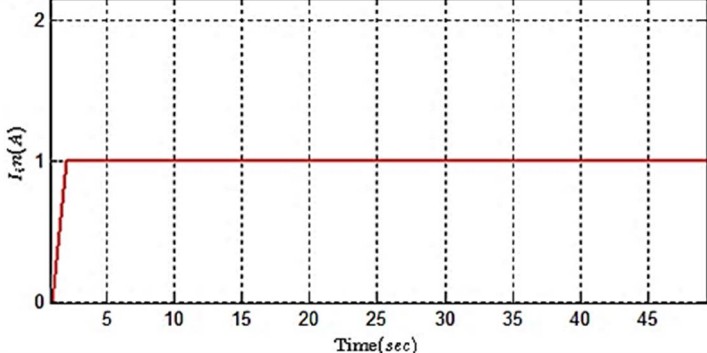

**Figure 3.** The second control input for the adaptive controller.

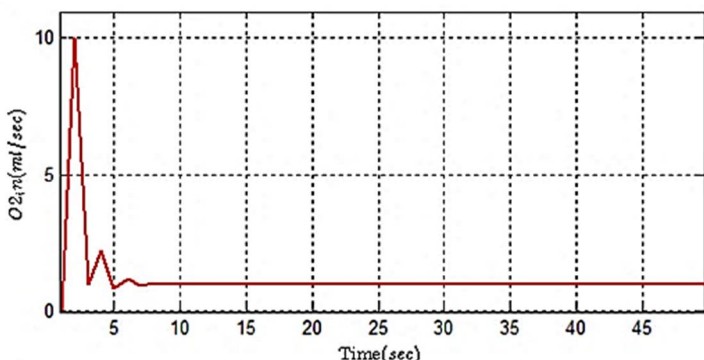

**Figure 4.** The third control input for the adaptive controller.

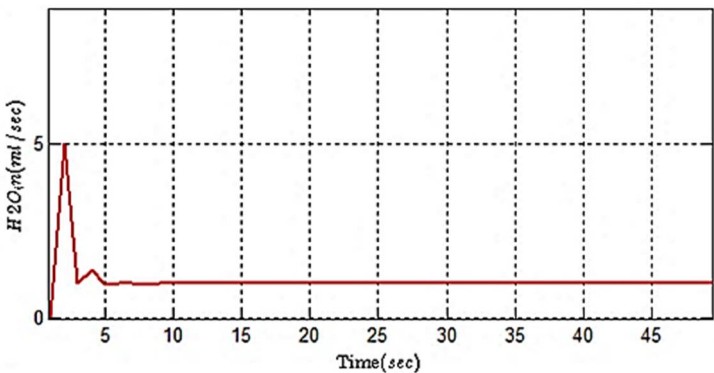

**Figure 5.** The fourth control input for the adaptive controller.

The control inputs are all bounded. The system states under the proposed adaptive controller are indicated in Figure 6.

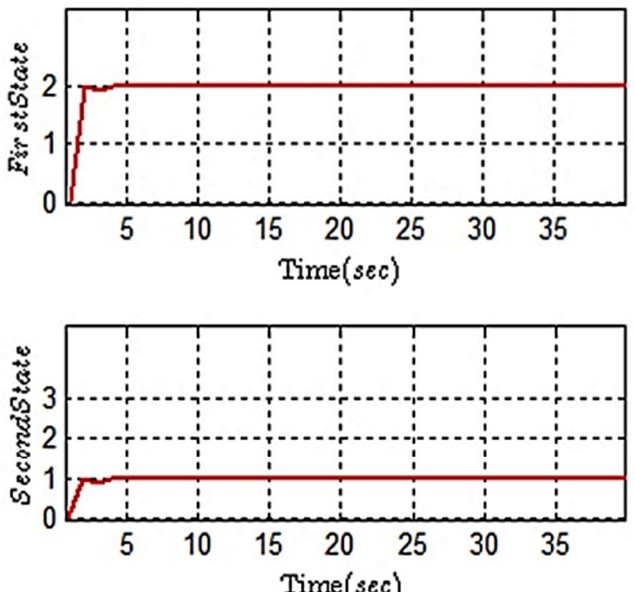

**Figure 6.** *Cont.*

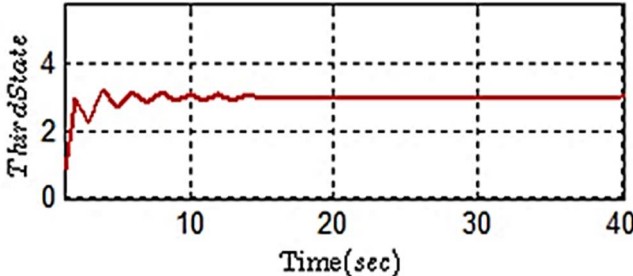

**Figure 6.** The states of a system under the proposed adaptive controller.

Figure 7 illustrates the control output of the PEMFC system under reference adaptive control. In this figure, the output of the adaptive controller is shown to rise to 6.2 V after the ripple voltage is applied, then decrease and stabilize at 2 V.

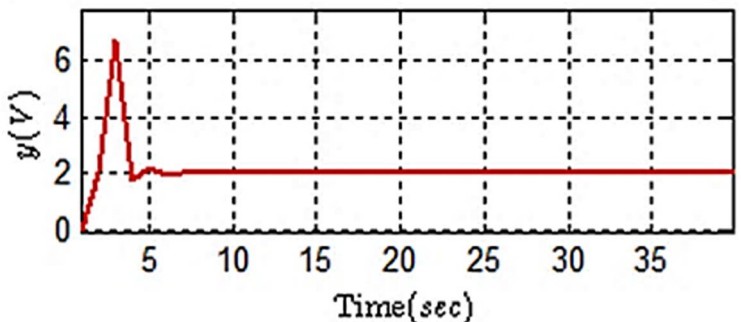

**Figure 7.** The output of the adaptive controller.

*4.2. Fuzzy Adaptive Controller Simulations*

The desired value of the control output was set to 2 V for a proper comparison of FAC and adaptive controller operation. Furthermore, the external 7reference signals were $x = [x_1, x_2, x_3]^T$ and $z = [x_1, x_2, x_3, v]^T$. It should be noted that $x_1$, $x_2$, and $x_3$ are defined over [−5, 5], and $v$ is defined over [−40, 40]. The fuzzy system defines four membership functions over the defined sets using the Gaussian function, as shown below:

$$\mu(x) = e^{\frac{(\chi - c)^2}{2\delta^2}} \tag{51}$$

where $\delta$ is the variance of the membership function and $c$ is its center. The initial values of all controller parameters were assumed to be zero. The system's output when an adaptive controller was applied is shown in Figure 7. Figure 8 depicts the system's state trajectory after applying the FAC. One of the most significant characteristics of any algorithm is its convergence speed. The suggested fuzzy adaptive controller converges quickly, as illustrated in Figure 9. Based on Equation (30), the final tracking error value is shown in Figure 8.

The following can be observed based on the Figures 8–10:

- In both presented methods, convergence of the tracking error to zero is guaranteed.
- The closed-loop system's stability is guaranteed in the presence of dynamic uncertainty.
- The fuzzy adaptive controller outperforms the adaptive controller in terms of tracking errors in the presence of uncertainties and disturbances.

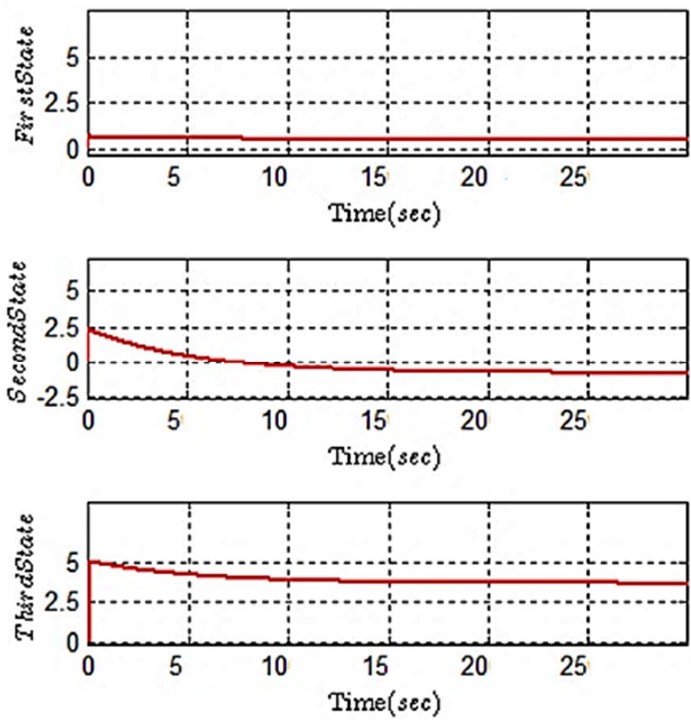

**Figure 8.** The system states under the proposed fuzzy adaptive controller.

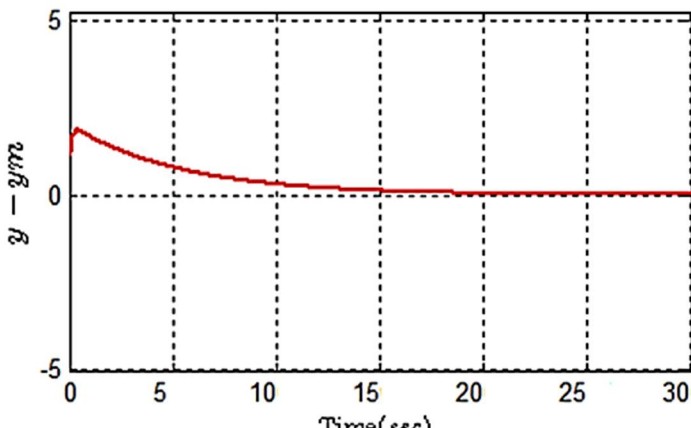

**Figure 9.** The tracking error of the proposed fuzzy adaptive controller.

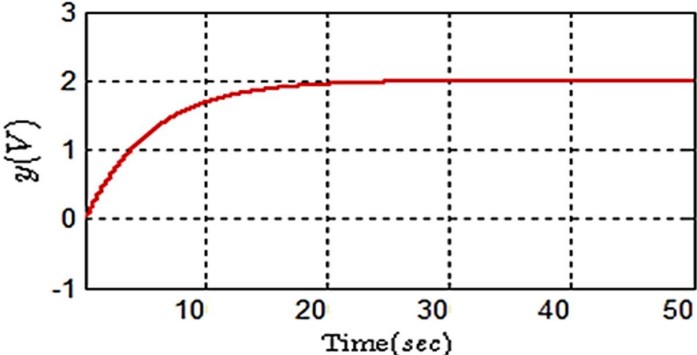

**Figure 10.** The control output of the fuzzy adaptive controller.

## 5. Conclusions

This paper presents both adaptive and fuzzy adaptive controllers for PEMFC systems with nonlinear models. The layout of the fuel cell is presumed to be understood in both plans, but the functions within this structure are completely unknown. These approaches show that such strategies can overcome both unknown dynamics and exogenous perturbations. An adaptation law ensures (1) closed-loop system stability, (2) asymptotic convergence of the tracking error to zero, and (3) the use of expert knowledge in the controller design. The fuzzy adaptive controller outperformed the adaptive controller in both tracking and dealing with uncertainties, but the adaptive controller is simpler to implement in practice than the fuzzy adaptive controller. These results confirm the superiority of the adaptive controller in terms of its transient response and demonstrate the effectiveness of the anticipated approaches as well. Future research could include an application of these methods to a nonlinear, nonaffine model of PEM fuel cell systems, as well as practical implementations of the suggested approach.

**Author Contributions:** R.G., methodology and validation; M.S., conceptualization and software; M.G., supervision and methodology; B.S.G., formal analysis and writing. All authors have read and agreed to the published version of the manuscript.

**Funding:** This research received no external funding.

**Institutional Review Board Statement:** Not applicable.

**Informed Consent Statement:** All authors have given their full consent for the submission and publishing of this manuscript in the MDPI journal *Sustainability*.

**Data Availability Statement:** Not applicable.

**Conflicts of Interest:** The authors declare no conflict of interest.

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
