# Peer review of "Design of a Fuzzy Adaptive Voltage Controller for a Nonlinear Polymer Electrolyte Membrane Fuel Cell with an Unknown Dynamical System"

_sustainability, doi:10.3390/su151813609_

Round 1

Reviewer 1 Report

This paper needs to be proofread thoroughly, and English needs to be improved significantly.

Author Response

TITLE: Design of a Fuzzy Adaptive Voltage Controller for a Nonlinear PEM Fuel Cell with an Unknown Dynamical System

Manuscript ID: 2497721

Authors: Reza Ghasemi, Mehdi Sedighi, Mostafa Ghasemi, Bita Sadat Ghazanfarpoor

Dear Editor

On behalf of all authors, I highly appreciate you and respected reviewers for the comments and correcting our mistake. We assure that their comments increased the level of our manuscript.

All the questions were answered carefully. The editor and reviewers' questions and concerns were shown by black colour; our response were shown by red colour. All the modification and changes in the manuscript has been shown by "red colour".

Once more, we thank and send our gratitude to you and reviewers.

Best regards

Dr. Mostafa Ghasemi

Email: mbaboli@su.edu.om, mostafghasemi@gmail.com

Reviewer #1:

The proposed work was intended to investigate “Fuzzy Adaptive Voltage Controller Design for Nonlinear Model of PEM Fuel Cell with Unknown Dynamical System”.

  • We would like to thank the reviewer for this encouraging evaluation. We thank you very much for providing many detailed suggestions and comments, most of which are excellent suggestions and we adopted. All changes in the manuscript are shown with red color. Some figures and tables in the revised manuscript have been modified.

Q1. The abstract can be rewritten to be more meaningful should clarify what is exactly proposed (the technical contribution) and how the proposed approach is validated.

Response:

  • Thank you for your comment. abstract is rewritten based on the proposed comments.

Q2. Literature review techniques have to be strengthened by including the issues in the current system and how the author proposes to overcome the same.

Response:

  • We appreciate for the reviewer’s comment. Previous studies focused on either affine nonlinear systems or fuzzy systems to approximate the unknown nonlinear functions of the PEM fuel cell, which resulted in both system structural limitations and increased computing volume. This paper presents a strategy for approximating controllers rather than nonlinear functions in order to reduce computing volume. The non-affine nonlinear PEM fuel cell is designated as a universal version of the nonlinear system in the presented approach.

Q3. There should be more literature in the work to align this work with previous studies.

Response:

  • Thanks a lot. We add it in the revised paper.

Q4. The references are not up-to-date.

Response:

  • Thank you for the fruitful comment. We update all the references.

Q5. Section 2 and Section 3 are cited from literature. What is the main purpose of writing basic Information in a manuscript?

Response:

  • Thanks a lot. We use this section to clarify the paper for both the chemical engineering and electrical engineering experts.

Q6. The novelty of this manuscript is not clear.

Response:

  • We appreciate for the reviewer’s comment. The novelty is added to the paper as follows:
  • In contrast to prior studies that focus on creating controllers for known dynamical models of PEM fuel cells, the suggested approach addresses a real-world scenario of a PEM fuel cell with unknown dynamics. An intelligent technique is identified in the suggested strategy to approximate the state-space model of fuel cells to manage unknown functions. The proposed methodology has the following advantages: 1) stability of the closed-loop system using Lyapunov, 2) robustness against external disturbances, 3) application of the FAC to a PEM fuel cell, 4) convergence of the tracking error to zero, and 5) overcoming both unknown dynamics and uncertainty in the system. The suggested system's main advantages are robustness, tracking of error convergence, and Lyapunov stability. Using the mathematical model of a multi-quadrotor system, this manuscript intends to demonstrate the responsiveness and fluency of the suggested technique.

Q7. The results should be compared and discussed with other papers published previously.

Response:

  • Thank you for the comment. Because of the differences in assumptions, dynamics, and disturbances in fuel cell modeling, it is difficult to apply the prior methods to the provided model. For these reasons, we developed and compared the classic adaptive technique discussed in prior references to our model.

Q8. More extensive simulations and more figures are needed.

Response:

  • Thank you for your comment. We add more figures in simulation results section.

Q9. This paper needs to be proofread thoroughly, and English needs to be improved significantly.

Response:

  • We appreciate for the reviewer’s comment. The English of the revised manuscript has been improved.

Q10. Future recommendations should be added in the conclusion

Response:

  • Thanks a lot. It is modified.

Reviewer 2 Report

In section 2.1, I think there is too much introductory education for Gibbs free energy, Nernst equation and voltage losses which are covered in undergraduate level physical chemistry courses. I don’t think it is necessary to include those equations and introductions for the sake of education in an article. However, in those introductions, in page 4 line 183, “According to [34], the following Nernst equation describes the reversible voltage of the cell”, in page 5 line 202, “which is described in [8] and [36], as below” : there are no denotations for the new parameters used in this article that are cited from the papers which makes readers hard to understand what the equations mean.

Page 4 line 168, there misses a plus sign before (Gf)H2 and makes the expression confusing.

Page 8 line 324, the format is not consistent with other math expressions.

Page 12, line 413, fig 10 should be fig 9?

The two models themselves look good to me and the one case simulation shows FAC has overall better performance than AC especially with uncertainties. However, the authors use too big a section in deriving the models and explaining the mathematical expressions which I think may be shortened and merged for many of the expressions.

In terms of model validation, the authors should strengthen their work by validating their controller design with real-world data or through experimental setups, in addition to the simulations they have performed. The one case data is not convincing enough.

I haven’t seen tracking error result with AC. Can you add it?

For all the figures especially with figure 2-10, please use higher quality pictures with higher resolution for better readability.

Author Response

TITLE: Design of a Fuzzy Adaptive Voltage Controller for a Nonlinear PEM Fuel Cell with an Unknown Dynamical System

Manuscript ID: 2497721

Authors: Reza Ghasemi, Mehdi Sedighi, Mostafa Ghasemi, Bita Sadat Ghazanfarpoor

Dear Editor

On behalf of all authors, I highly appreciate you and respected reviewers for the comments and correcting our mistake. We assure that their comments increased the level of our manuscript.

All the questions were answered carefully. The editor and reviewers' questions and concerns were shown by black colour; our response were shown by red colour. All the modification and changes in the manuscript has been shown by "red colour".

Once more, we thank and send our gratitude to you and reviewers.

Best regards

Dr. Mostafa Ghasemi

Email: mbaboli@su.edu.om, mostafghasemi@gmail.com

Reviewer #2:

In section 2.1, I think there is too much introductory education for Gibbs free energy, Nernst equation and voltage losses which are covered in undergraduate level physical chemistry courses. I don’t think it is necessary to include those equations and introductions for the sake of education in an article. However, in those introductions, in page 4 line 183, “According to [34], the following Nernst equation describes the reversible voltage of the cell”, in page 5 line 202, “which is described in [8] and [36], as below” : there are no denotations for the new parameters used in this article that are cited from the papers which makes readers hard to understand what the equations mean.

  • We would like to thank the reviewer for this encouraging evaluation. We thank you very much for providing many detailed suggestions and comments, most of which are excellent suggestions and we adopted. All changes in the manuscript are shown with red color. Some figures and tables in the revised manuscript have been modified.

Q1. Page 4 line 168, there misses a plus sign before (Gf)H2 and makes the expression confusing.

Response:

  • We appreciate for the reviewer’s comment. It is modified.

Q2. Page 8 line 324, the format is not consistent with other math expressions.

Response:

  • Thanks a lot. We modified it.

Q3. Page 12, line 413, fig 10 should be fig 9?

Response:

  • Thanks a lot. It is modified.

Q4. The two models themselves look good to me and the one case simulation shows FAC has overall better performance than AC especially with uncertainties. However, the authors use too big a section in deriving the models and explaining the mathematical expressions which I think may be shortened and merged for many of the expressions.

Response:

  • Thank you for the fruitful comment. We merged the modeling portion, but explain the dynamics to make the study more understandable for both chemical engineering and electrical engineering researchers.

Q5. In terms of model validation, the authors should strengthen their work by validating their controller design with real-world data or through experimental setups, in addition to the simulations they have performed. The one case data is not convincing enough.

Response:

  • Thanks a lot. An extension of these methods to nonlinear non-affine model of PEM fuel cell systems and practical implementations of proposed approach can be considered in future studies.

Q6. I haven’t seen tracking error result with AC. Can you add it?

Response:

  • Thanks a lot. It is modified in the in simulation result section.

Q7. For all the figures especially with figure 2-10, please use higher quality pictures with higher resolution for better readability.

Response:

  • Thank you for the attention. We improved the quality of the figures.

Reviewer 3 Report

The article entitled „Fuzzy Adaptive Voltage Controller Design for Nonlinear Model of PEM Fuel Cell with Unknown Dynamical System” is an interesting paper. The methods used are suitable for the intended purpose, and the results are notable. The physical derivation, which supports the claims with its explanatory power, is the strength of the article.

Acceptable

Author Response

TITLE: Design of a Fuzzy Adaptive Voltage Controller for a Nonlinear PEM Fuel Cell with an Unknown Dynamical System

Manuscript ID: 2497721

Authors: Reza Ghasemi, Mehdi Sedighi, Mostafa Ghasemi, Bita Sadat Ghazanfarpoor

Dear Editor

On behalf of all authors, I highly appreciate you and respected reviewers for the comments and correcting our mistake. We assure that their comments increased the level of our manuscript.

All the questions were answered carefully. The editor and reviewers' questions and concerns were shown by black colour; our response were shown by red colour. All the modification and changes in the manuscript has been shown by "red colour".

Once more, we thank and send our gratitude to you and reviewers.

Best regards

Dr. Mostafa Ghasemi

Email: mbaboli@su.edu.om, mostafghasemi@gmail.com

Reviewer #3:

The article entitled ,Fuzzy Adaptive Voltage Controller Design for Nonlinear Model of PEM Fuel Cell with Unknown Dynamical System” is an interesting paper. The methods used are suitable for the intended purpose, and the results are notable. The physical derivation, which supports the claims with its explanatory power, is the strength of the article.

  • We would like to thank the reviewer for this encouraging evaluation. We thank you very much for providing many detailed suggestions and comments, most of which are excellent suggestions and we adopted. All changes in the manuscript are shown with red color. Some figures and tables in the revised manuscript have been modified.

Reviewer 4 Report

The paper reported a Fuzzy Adaptive Controller (FAC) for improving the efficiency and stability of fuel cells. The authors designed an intelligent observer to estimate the state-space model of fuel cells to handle unknown functions. However, the paper lacks a clear and concise central idea. The introduction does not provide a strong enough background to justify the need for the study, and the research objectives are not well-defined. The literature review and main results appear to be insufficient, and the authors have not made a compelling argument for the originality of their work.

1. Introduction is very redundant and the logical flow of it is very confusing, especially the first paragraph about PEM fuel cell. It looks like several advantages and disadvantages of PEMFC were combined randomly which makes readers hard to follow.

2. Page 2, line 68-70, repeated sentences as above.

3. Page 2, line 74, full name of PID control is missing.

3. Page 2, line 86-89, what is the relationship between the lifetime and electrical efficiency of fuel cell and this study? There seems to be no connection.

4. Page 2, line 91, no reference was provided regarding FAC here.

5. Page 3, line 133, Following is an outline of the rest of the paper? What is the outline of the paper? Why is the outline shown in a research paper?

6. Page 3, line 141, “fuel cell small-signal models are examined in more detail [2] through [10].”

References can’t be used like this.

7. Page 4, open circuit voltage of PEMFC is not related to the study. It’s not necessary to show this content that we can find in every textbook.

8. All the figures and labels look very blurry, which are hard to read.

9. Conclusions provide closure for the reader while reminding the reader of the contents and importance of the paper. It accomplishes this by stepping back from the specifics to view the bigger picture of the document. However, the conclusions of this paper don’t provide these points and just show what they did.

10. Many typos can be found in the paper. It is noted that your manuscript needs careful editing by someone with expertise in technical English editing paying particular attention to English grammar, spelling, and sentence structure so that the goals and results of the study are clear to the reader.

Many typos can be found in the paper. It is noted that your manuscript needs careful editing by someone with expertise in technical English editing paying particular attention to English grammar, spelling, and sentence structure so that the goals and results of the study are clear to the reader.

Author Response

TITLE: Design of a Fuzzy Adaptive Voltage Controller for a Nonlinear PEM Fuel Cell with an Unknown Dynamical System

Manuscript ID: 2497721

Authors: Reza Ghasemi, Mehdi Sedighi, Mostafa Ghasemi, Bita Sadat Ghazanfarpoor

Dear Editor

On behalf of all authors, I highly appreciate you and respected reviewers for the comments and correcting our mistake. We assure that their comments increased the level of our manuscript.

All the questions were answered carefully. The editor and reviewers' questions and concerns were shown by black colour; our response were shown by red colour. All the modification and changes in the manuscript has been shown by "red colour".

Once more, we thank and send our gratitude to you and reviewers.

Best regards

Dr. Mostafa Ghasemi

Email: mbaboli@su.edu.om, mostafghasemi@gmail.com

Reviewer #4:

The paper reported a Fuzzy Adaptive Controller (FAC) for improving the efficiency and stability of fuel cells. The authors designed an intelligent observer to estimate the state-space model of fuel cells to handle unknown functions. However, the paper lacks a clear and concise central idea. The introduction does not provide a strong enough background to justify the need for the study, and the research objectives are not well-defined. The literature review and main results appear to be insufficient, and the authors have not made a compelling argument for the originality of their work.

  • We would like to thank the reviewer for this encouraging evaluation. We thank you very much for providing many detailed suggestions and comments, most of which are excellent suggestions and we adopted. All changes in the manuscript are shown with red color. Some figures and tables in the revised manuscript have been modified.

Q1. Introduction is very redundant and the logical flow of it is very confusing, especially the first paragraph about PEM fuel cell. It looks like several advantages and disadvantages of PEMFC were combined randomly which makes readers hard to follow.

Response:

  • Thanks a lot. We revised the introduction section.

Q2. Page 2, line 68-70, repeated sentences as above.

Response:

  • Thanks for your consideration. It is modified.

Q3. Page 2, line 74, full name of PID control is missing.

Response:

  • Thanks a lot. Proportional- Integral-Derivative (PID) is added in paper.

Q4. Page 2, line 86-89, what is the relationship between the lifetime and electrical efficiency of fuel cell and this study? There seems to be no connection.

Response:

  • Many thanks to the honorable reviewer. There is no connection between lifetime and electrical efficiency. We mean that in majority of the presented approaches, it is assumed that " PEMFCs are estimated to have a lifetime of 40000 hours, and their electrical efficiency is approximately 40-50% ".

Q5. Page 2, line 91, no reference was provided regarding FAC here.

Response:

  • Thank you very much. It is modified.

Q6. Page 3, line 133, Following is an outline of the rest of the paper? What is the outline of the paper? Why is the outline shown in a research paper?

Response:

  • Thanks a lot. It is modified.

Q7. Page 3, line 141, “fuel cell small-signal models are examined in more detail [2] through [10].” References can’t be used like this.

Response:

  • Thank you for your careful check. It is modified.

Q8. Page 4, open circuit voltage of PEMFC is not related to the study. It’s not necessary to show this content that we can find in every textbook.

Response:

Thank you for your perspective. In the next subsection, we need these formulations to clarify the paper.

Q9. All the figures and labels look very blurry, which are hard to read.

Response:

  • Thank you for your careful check. We revised the figures in the paper.

Q10. Conclusions provide closure for the reader while reminding the reader of the contents and importance of the paper. It accomplishes this by stepping back from the specifics to view the bigger picture of the document. However, the conclusions of this paper don’t provide these points and just show what they did.

Response:

  • Thanks a lot. We rewrite the conclusion again based on your comments.

Q11. Many typos can be found in the paper. It is noted that your manuscript needs careful editing by someone with expertise in technical English editing paying particular attention to English grammar, spelling, and sentence structure so that the goals and results of the study are clear to the reader.

Response:

  • Thanks for your remarks. We have revised whole manuscript carefully and tried to avoid any grammar on syntax error. In addition, we have asked several colleagues who are skilled author of English language papers to check the English. We believe that the English is now acceptable for the review process.

Round 2

Reviewer 1 Report

The authors address all my concerns. No further comments. 

Minor editing of English language required

Reviewer 2 Report

The paper is in a better shape now and can be accepted.

Reviewer 4 Report

All the issues were addressed. The present form is accepted.